# First Record of Cetacean Killed in an Artisanal Fish Aggregating Device in the Mediterranean Sea

**DOI:** 10.3390/ani13152524

**Published:** 2023-08-04

**Authors:** Valerio Manfrini, Caterina Maria Fortuna, Cristiano Cocumelli

**Affiliations:** 1Independent Researcher, 44022 Comacchio, Italy; 2Italian Institute for Environmental Protection and Research, 00144 Rome, Italy; 3Istituto Zooprofilattico Sperimentale del Lazio e della Toscana “M. Aleandri” [Public Health Institution], 00178 Rome, Italy; cristiano.cocumelli@izslt.it

**Keywords:** fish aggregating device (FAD), bycatch, striped dolphin, Mediterranean Sea

## Abstract

**Simple Summary:**

Fish Aggregating Devices (FADs) are floating structures used to create shadows and aggregate pelagic fish. In tropical and subtropical waters, the dangerous impact of these structures, due to high levels of bycatch, is well known. We report the first case of a cetacean killed by a FAD and stranded on the coast of Lazio, Italy (central Tyrrhenian Sea). The postmortem evaluation, the difference in degradation (necrosis) between the tail and peduncle, and the marked malnutrition confirm that this specimen was still alive at the time of entanglement. Although this is the first verified case of a cetacean entangled in a FAD in Mediterranean waters, given the widespread use of illegal versions of these devices, the extent of the problem may be greatly underestimated.

**Abstract:**

Fish Aggregating Devices (FADs) are anchored floating structures often made with cheap scrapped materials and used to aggregate pelagic fish species under their artificial shadows. Globally, the dangerous impact of FADs is well known. They pose a severe threat as a source of bycatch, as a danger to navigation, and with their high potential to become marine litter. Unintended entanglement and consequent mortality in FADs of vulnerable (e.g., sharks, sea turtles, and cetaceans) and commercial species is a serious concern for several international inter-governmental bodies (e.g., EU, GFCM, and IWC). This work describes the first case of a cetacean, a striped dolphin (*Stenella coeruleoalba*), entangled in a FAD in the Mediterranean Sea. A young male of striped dolphins was found dead along the coast of Lazio (central Tyrrhenian Sea) with its peduncle entangled in typical debris from illegal/artisanal FADs (i.e., a nylon rope, teared gardening plastic sheets, bush branches, and scrapped empty plastic bottles). Although this is the first confirmed case of a cetacean entangled in a FAD in Mediterranean waters, given the extent of the deployment of anchored FADs, the scale of this type of interaction with protected species might be seriously underestimated. Therefore, actions and monitoring need to be implemented urgently to effectively protect and conserve marine biodiversity.

## 1. Introduction

FADs are floating items strategically positioned to attract fish and are composed by a surface portion connected by a rope to an anchoring system [1]. They have been used by fishermen in the Mediterranean Sea for thousands of years [2]. Until the end of the 1990s, from August to December, in Mediterranean waters around Malta, Tunisia, Sicily, and Mallorca, over 2300 boats were using anchored FADs to target dolphinfish (*Coryphaena hippurus*) [3]. FADs are usually anchored in specific locations, ranging from very coastal to offshore waters (over depths up to 3500 m) [4]. Within the Mediterranean Sea, they are usually made of various floating scrap materials [4]. Even though these FADs are targeting dolphinfish using special encircling nets, pilotfish (*Naucrates ductor*) and greater amberjack (*Seriola dumerili*) are also common valuable bycatch species [3].

In times of industrial fishery expansion and technological advances, this practice has been acquired globally on a massive scale and is particularly linked to industrial tuna fishery [2]. In some regions of the world, FADs have an important impact on a wide range of vulnerable, protected, and commercial species, many of which are caught accidentally during fishing operations or become entangled in the elements of these devices [5,6]. These include sea turtles, sharks, and many juvenile fish and fish species that are not the targeted catch of these fisheries [5,6].

The European Union (EU) and the General Fisheries Commission for the Mediterranean (GFCM) share fishery policy principles on the need to ensure a sustainable ecosystem-based management framework for fisheries, which guarantees adequate protection of vulnerable species and sensitive habitats [7,8]. Fishing practices negatively affecting the whole marine ecosystem or elements of it need to be carefully managed or even prohibited [7,8].

Due to the globally recognized impact of the use of FADs on vulnerable, protected, and commercial species, as well as on marine ecosystems, and the danger that they pose to navigation, the GFCM started raising awareness about these fishing devices almost 20 years ago [9]. More recently, the GFCM has established a set of production and deployment standards [10] that, if not followed, make FADs in the Mediterranean Sea illegal. These standards specify, inter alia, that FADs are allowed only if they meet all the following standards:(i)They target the common dolphinfish, and are only for vessels that hold a valid fishing authorization issued by the competent authorities;(ii)All their elements are made of biodegradable materials;(iii)Their surface elements are made of ‘*material that involve minimal risk of entangling non-target species, especially vulnerable species, or affecting other vessels*’;(iv)They are reliably locatable once deployed;(v)Their main rope is equipped with ‘*an appropriate number of counterweights*’ to make it sink to the sea bottom if the surface elements drift away;(vi)They are properly marked to be readily identified (i.e., visible registration numbers of fishing vessels);(vii)Vessels deploying FADs must record and report back to their national authorities all information on their fishing activities, including any loss of FADs with their latest position.

In addition, according to the GFCM recommendations, national fishery authorities must ensure that FADs are regularly maintained and replaced as necessary.

Currently in the GFCM registry [11] of ‘Authorized vessels using Fish Aggregating Devices for the exploitation of common dolphinfish in the Mediterranean (GSAs 1 to 27)’, there are 404 registered vessels from three Mediterranean countries: Italy (261), Malta (104), and Spain (39). Most of these 404 vessels belong to the fleet segments VL6-12 (302 vessels) and VL12-24 (86 vessels). Considering that vessels deploy an average of 30 to 100 FADs [3,12], it can be estimated that there are 12,000–40,000 potentially legal FADs around the Mediterranean Sea [3,12] if they comply with the above listed standards (i–vii). However, numbers could be much higher (e.g., 70,000–230,000) if one considers the original census of 2300 vessels using this fishing device and that Tunisian fishers, previously known for deploying FADs, are not present in the GFCM registry [3,11,13].

These numbers make this fishing practice a massive potential source of (a) incidental captures of protected and vulnerable species (including fish) [5,6], (b) marine litter [1], and (c) risk for navigation [2,14].

In addition, within the Mediterranean, most of the existing artisanal FADs are not used in legitimate fishing operations either because they are not conducted by authorized vessels, or because they do not meet any (or most) of the GFCM recommendations linked to FAD design to minimize the risk of entanglement for vulnerable species, plastic pollution, and navigation risk [2,14]. Between 2017 and 2022, a study carried out in the southern Tyrrhenian Sea recovered 1739 illegal FADs, which were deployed in waters of depths ranging between 526 and 3518 m [4].

Cetaceans in Italy are protected from killing and human disturbance under national and European legislation. In particular, by the Italian Law 157/1992 [15], the EU Council Directive 92/43/EEC [16], and the Italian Presidential Decree 357/1997 [17] and subsequent modifications.

This work aims to describe the first case of a cetacean killed by a FAD in the Mediterranean Sea (Italy) to raise awareness on this likely underestimated threat to cetacean populations in this region.

## 2. Materials and Methods

The Italian Stranding Network is a complex system that relies on cooperation between the coastguard, local health authorities, and National Center for Diagnostic Investigations of Stranded Marine Mammals (*Centro di referenza nazionale per le indagini diagnostiche sui mammiferi marini spiaggiati*, CReDiMa) that coordinates the *Istituti Zooprofilattici Sperimentali* (Public Health Institutes) distributed across national territory, universities, the Natural History Museum of Milan, and volunteers of associations/organizations such as the *Centro Studi Cetacei* [18,19]. It operates in line with Agreement on the Conservation of Cetaceans of the Black Sea, Mediterranean Sea, and contiguous Atlantic area (ACCOBAMS) guidelines on this matter [20].

Concerning this specific event, we collected all evaluations and body measurements on site except for tail width and peduncle circumference due to their corruption. The postmortem examination was performed by the staff of the *Istituto Zooprofilattico Sperimentale del Lazio e della Toscana*, located in Rome, according to the ACCOBAMS guide on ‘*Best practice on cetacean postmortem investigation and tissue sampling*’ [21].

## 3. Results

This note concerns the stranding of a striped dolphin (*Stenella coeruleoalba*) (Meyen, 1833) (*Cetartiodactyla*, *Delphinidae*) found on 29 November 2019, along the west coast of the central Tyrrhenian Sea (Marina di Ardea) near Rome (Figure 1). The specimen was a 176 cm long male, weighing 40 kg; hence, classified as young [22].

The dolphin was tightly entangled in a nylon rope attached to various debris, including remains of a long black nylon rope attached to remains of a green plastic sheets (usually used for gardening), two large empty 5 L plastic bottles, and some large bush branches. These were identified as typical components of medium-size artisanal and illegal FADs (Figure 2A).

The rope was tightly wrapped around the end of its peduncle, in proximity of the tail, and had implanted deep into its tissues, leading to visible necrosis in the surrounding tissues (part of the peduncle and the whole tail, see Figure 2B). Externally, this individual was in a moderate decomposition (code 3 out of 5), while internally, organs were in an advanced decomposition (code 4) [21,23]. Furthermore, it had poor nutritional status, evidenced by a dorsal blubber thickness of 6 mm.

The postmortem examination showed an empty gastrointestinal system, with no food remaining. Regarding the tail, the state of decomposition of the tissues was severe and much worse than the rest of the body. The almost complete loss of firmness, pallor, and degree of tissue necrosis was considered a consequence of the stricture due to the better conservation of the tissues cranial at the nylon knot and the deepening of the rope in the soft blubber tissues. These features confirm that the injuries occurred when the animal was still alive, and they were unlikely generated instantaneously. PCR tests detected the presence of *Herpesviruses* in brain, spleen, and lung tissue. Histological examination was attempted only on the lung to highlight signs of drowning/asphyxia, but the poor state of conservation of all internal organs did not allow a clear result to be obtained.

A few days later, just 9 km north from the stranding location (Figure 2), a floating part of a small illegal FAD was found (Figure 3).

## 4. Discussion

Opportunistic reports of cetacean entanglements in FADs exist for sperm whales (*Physeter macrocephalus*) in Guadalupe [24] and they are suspected in the whole Caribbean region for various other cetacean species [25]. Sperm whales and sei whales (*Balaenoptera borealis*) are known to interact with FAD structures, increasing the likelihood of accident [25]. In addition, two curious accounts of a sperm and a humpback whale (*Megaptera novaeangliae*) opportunistically used as FADs themselves have been described in industrial tuna and shark fisheries in Ecuador [26]. This fishery, as others in South American countries, is known to use all sorts of floating objects to attract and capture pelagic fish, often in combination with marine mammal as baits [26].

According to the existing literature, this is the first case of a cetacean entangled in a FAD in the Mediterranean Sea, but the extent of the problem may be severely underestimated. This could be due to both a lack of specific forensic protocols to record such occurrences and a wide-spread use of illegal versions of FADs in the Mediterranean region.

The age of the animal was inferred from the existing literature for the region, where adult size is about 200 cm for a weight of 80–120 kg [22,27].

The evidence gathered by the postmortem examination—particularly on the necrotic tissue around the peduncle, the emaciated body, and the empty stomach—suggests that, after becoming entangled into the rope of an artisanal FAD, this animal survived for some time dragging parts of this illegal FAD.

In Italy, FADs are mainly used in southern regions [1,3,4]; however, our accounts of an entangled dolphin and an abandoned small FAD on the shore of central Italy, may suggest that this practice is used in other Italian regions. The official numbers reported by the GFCM registry and official national registries seem also severe underestimates of the actual the situation, given the evidence of a widespread traditional use of FADs, e.g., [1,3,4,12,28]. It is also unclear why Tunisia is not reporting any activity linked to the use of FADs, since Tunisian fishermen were historically known to use such devices [3].

Past experimental studies demonstrating that traditional FADs made with palm leaves provided a much less efficient attractant device for some very valuable catch (i.e., the greater amberjack, *Seriola dumerili*) compared to floats made of synthetic material [29] might have incentivized the bad habit to take advantage of much cheaper and readily available scrapped non-biodegradable plastic materials. It has been ascertained that, within the Mediterranean Sea, FADs usually contain a large proportion of non-biodegradable materials, including discarded detergent plastic bottles (5 L), parts of tires, polystyrene slabs often wrapped in plastic sheets, polyethylene, nylon ropes, and anchoring limestone or concrete blocks [1,4]. In our opinion, the fact that these fishing devices are made of scrapped material does not incentivize fishers for their recovery once they are lost. Several authors suggested that their loss or abandonment equates to ‘dumping’, contravening international marine pollution law [1,14]. In particular, their systematic loss would breach the Annex V of International Convention for the Prevention of Pollution from Ships (MARPOL) [1,14]. In any case, these artisanal FADs are clearly not in line with established GFCM and EU fishery standards [10], which encourage preventing entanglements of non-target species, minimizing pollution and losses, and maximizing recovery of lost FADs [14].

## 5. Conclusions

This paper provides a record of a likely highly underestimated problem that affects cetaceans and other protected species. Depending on the status of the carcasses, many of such entanglements might go undetected. In addition, there is not yet a protocol to systematically gather evidence on stranding events that would allow for an estimation of the rate of such an interaction. Even though the Mediterranean striped dolphin is listed as Least Concern by the IUCN [30], this species is strictly protected under the Annex IV of the EU Habitat Directives. The same is true for the loggerhead sea turtle (*Caretta caretta*), also listed in Annex IV of the EU Habitat Directives, which is known to be caught incidentally by FADs in the southern Tyrrhenian Sea [31], but the severity of this impact is unknown. Under European legislation, all species listed in Annex IV of the EU Habitat Directives needs to be protected from human-induced disturbance and mortality, and monitoring programs need to be in place to estimate bycatch rates (see article 12 paragraph 4 of the EU Council Directive 92/43/EEC [16]). The described scenario is particularly worrisome considering that over 82% of the anchored FADs deployed annually across the world are in the Mediterranean [28] and that the estimated amount of lost or abandoned FADs is enormous [1,4]. In 2018, the International Whaling Commission (IWC) expressed concern at the rapid increase in deployment of FADs in many parts of the world and the links with entanglement of cetaceans [25]. We share the IWC concern and call for the urgent enforcement of existing GFCM and EU fishery regulations in regard to FADs deployment, especially in the Mediterranean Sea.

## Figures and Tables

**Figure 1 animals-13-02524-f001:**
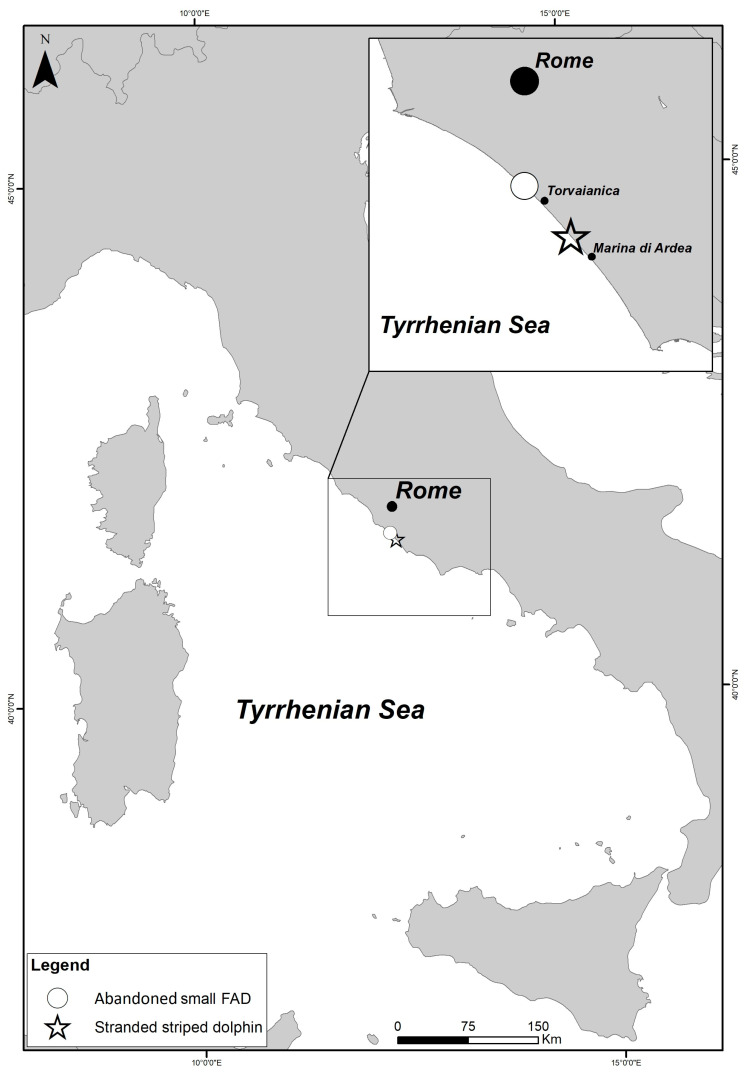
Location of the stranding (✩) Marina di Ardea (Rome) (41.5853 N, 12.4995 E) and location of parts of another illegal FAD (○) found abandoned on the beach of Torvaianica (Rome) (41.647648 N, 12.427815 E).

**Figure 2 animals-13-02524-f002:**
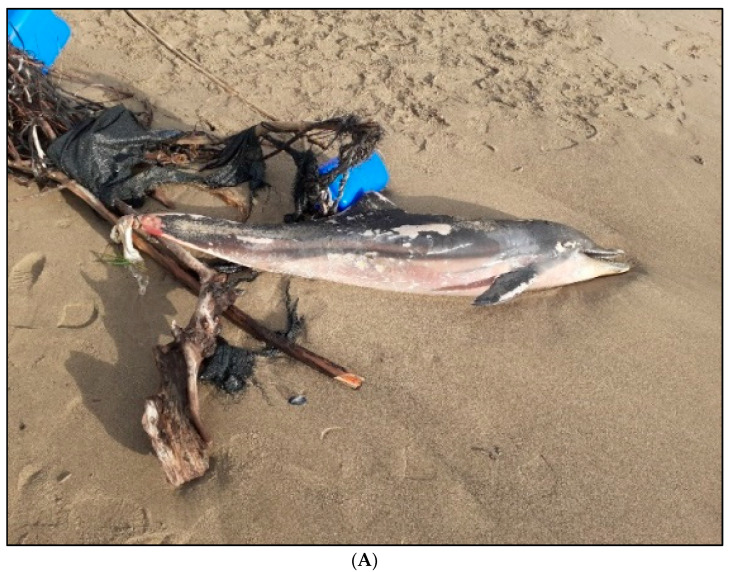
Photographic evidence by V. Manfrini: (**A**) beached striped dolphin (*Stenella coeruleoalba*) and remains of an illegal FAD, i.e., two scrapped 5 L plastic bottles with no vessel registration numbers, long nylon ropes, a teared dark-green plastic sheet, and the remainder of a bush branch; (**B**) detail on the peduncle (large hematomas) and tail (necrotic tissue) of the striped dolphin with the FAD’s parts.

**Figure 3 animals-13-02524-f003:**
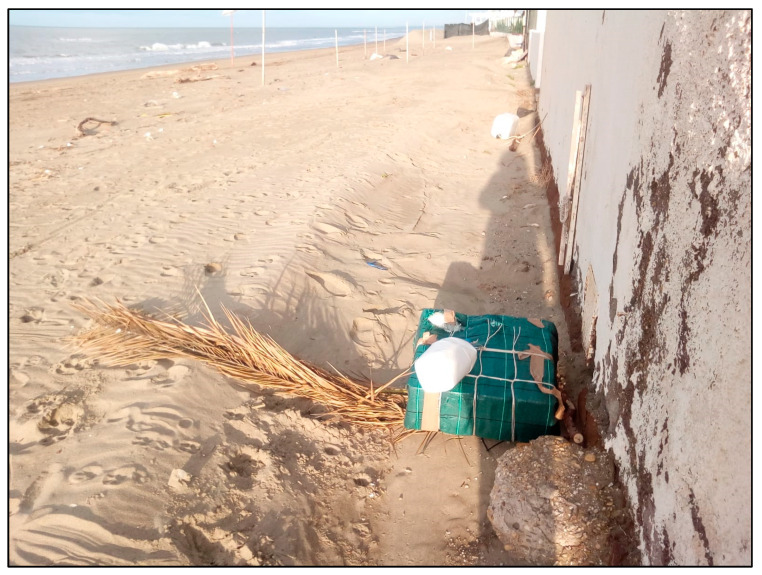
An example of a small artisanal FAD photographed 9 km north the location of the stranded striped dolphin. Note the green plastic sheet of the same kind found with the stranded striped dolphin. (Photo by E. Santini).

## Data Availability

‘Not applicable’ here; all data are reported in the manuscript.

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
