# Peer review of "First Record of Cetacean Killed in an Artisanal Fish Aggregating Device in the Mediterranean Sea"

_animals, 2023, doi:10.3390/ani13152524_

Round 1
Reviewer 1 Report
Review of animals-2471682 “First record of cetacean killed in an artisanal Fish Aggregating Device in Italian waters”
General comment:
Manfrini et al. reported the first case of a striped dolphin killed by fish aggregating device in the central Tyrrhenian Sea. Manfrini et al.’s results suggested that painful death for the cetacean due to the entanglement and consequent mortality in FAD. Finally Manfrini et al. discussed the importance of actions and monitoring that need to be implemented to effectively protect and conserve marine biodiversity.
The introduction, methodology, results and discussion are generally well presented. However, the manuscript needs major changes before being accepted. Details are listed below:
Summary/Abstract
Line 13: please, use lower case for Central
Line 14: please, replace with ‘the difference in degradation (necrosis) between the tail and peduncle’, otherwise clarify the type of degradation of the peduncle
Lines 19: please replace with ‘shadows’
Lines 26-27: please expand more the results (see the simple summary)
Introduction
Line 34: please, replace with FADs
Line 37: please, remove (FAD)
Line 38-39: please, add space between number and m
Line 47: please, add a reference or, if applicable, move the reference at the end of the sentence
Line 50: please, provide full name for EU and mention correctly the UN FAO GFCM
Lines 55-56: please, replace with ‘vulnerable species and ecosystems’, as FADs is affecting more directly species and indirectly ecosystems
Line 71: please, use lowercase for authorities
Lines 87-91: please, add reference
Lines 95-96: please, add reference/links to each legal document
Materials and Methods
Lines 106: please, provide full name for ACCOBAMS
Line 109-111: please, add description of the methodology used for the post-mortem examination
Line 114: please, remove multiple word spacing before (Fig.1)
Figure 1: please, re-size and make readable all components (e.g. coordinates, names etc). Why 2nd FAD? Where is the 1st one? It is not mentioned in the text. Finally the FAD looks located on the coast. Please, correct accordingly and increase the resolution of the figure
Results
Line 120: is 176 cm a young individual? Please describe more the beached cetacean
Lines 120-121: code 3/4? Please,. add meaning and reference
Lines 128-129: please, indicate the described FAD in the figure
Lines 136-137: please, add the meaning /interpretation of the detection
Discussion
Lines 152-154: please, ad references
Lines 177-179: please add reference or make clear that it is an author’s statement
Line 179: please use lower case for Authors
Line 182: please, provide full name for MARPOL
Line 184: please, remove and or complete the sentence.
Line 190: please, replace with ‘this species is…’, and add reference for EU Habitat Directives
Line 192: please, rewording ‘it is known to be bycaught I FADs’
Line 196: please, add reference for EU Council Directive 92/43/EEC)
Line 199: please, provide full name for IWC
Minor editing of English language required (see comments above)
Reviewer 2 Report
After revision of this manuscript, I really do not understand the main point of the document, the authors insist to say this is the first report of a cetacean death in a Fish Aggregating Device FAD. But authors dedicated many lines of the manuscript to describe de artisanal FAD the illegal FAD and the rules for FAD.
On the other hand, the striped dolphin was found killed on 29th November 2019 near Rome. The question that rises here is, there is no previous report or legal investigation by authorities? It is not clear the real intentions of the manuscript. If the intentions is about the cruelty on how the dolphin died, the author also failed
Reviewer 3 Report
This manuscript shows a very concerning case of a possible lethal interaction between Fish Aggregating Devices and a striped dolphin.
However, I think there are several issues to address in order to present a more organised and clear paper.
Here are some specific comments:
Is it necessary to present this ‘Simple Summary’?
Line 10 ‘to create shadows’?
Lines 11-12 how these structures produce bycatch, if they only ‘create shadows’? Please clarify
Line 12 ‘killed by ‘ a ‘fish aggregating device’.
Line 13 ‘coast’ OF ‘Rome’.
Line 19-20 this sentence has no meaning. Do you mean IS well known?
Lines 22-23 concern two times too close
Line 24 please the first time you use acronyms, you have to spell the whole word (e.g. what does GFCM stand for?)
Lines 26-27 a painful death? How did you measured that? Please clarify
Lines 27-28 Is it the first report of a cetacean killed by a FAD in the Mediterranean, why say that it is the first in Italy that only has Mediterranean coastline?
Line 29 actions like what? No suggestions?
Line 36 please, Mallorca
Lines 33-42 During the start of the introduction you never got to really describe how a FAD is made. Is it a floating pallet with other object attached? Please describe examples of FADs.
Line 55 what are essential fish? Essential to whom? Please clarify
Lines 82-83 where these estimates came from? There is a big gap between 70k and 230k.
Lines 84-86 why there is a reference only for one of the impacts?
Lines 87-93 Is that the reference for such a strong sentence? How comes that ‘most’ are not used in legitimate fishing, then you reference 1739 illegal FADs but you have previously talked about 40k registered FADs.
Line 94 ‘protected from disturbance’? please explain
Lines 94-96 That final sentence comes suddenly and makes an abrupt end to the introduction. You have not mentioned your objectives, this is some major flaw.
Materials and Methods have sentences that belong to the Introduction section. Moreover, they are too succinct, please expand on the necropsy work.
Line 121 how do you measure the poor nutritional status? Please clarify what made you annotate that.
Line 121-122 This whole sentence changes your paper, please be clear and specify in the title that this is a PROBABLE or POTENTIAL killing by a FAD. Also, discuss the possibility that this event happened differently. It most likely happened as you describe, but I think you should suggest it, not affirm it.
Line 138 ‘Unfortunately’ here seems more appropriate for a dialogue conversation.
Line 184 ‘and’?
Lines 185-188 this is a long sentence, maybe try to split it?
Line 192 ‘I FAD’?
Line 193 ‘at unknown levels’?
Line 212 the The Italian Stranding Network receives no funding?
Line 213 maybe it reads better if you state that all the data is shown within the paper.
Line 221 On a personal note, these are very nice words for the late Gianni Pavan, well done.
Please review the English, some examples have been pointed out in the specific comments on your manuscript.
Round 2
Reviewer 1 Report
Review of animals-2471682_v2 “First record of cetacean killed in an artisanal Fish Aggregating Device in Mediterranean Sea”
General comment:
The manuscript needs few minor changes before being accepted. Details are listed below:
Title/Summary/Abstract
Line 3: please replace with “in the Mediterranean Sea” or “Mediterranean waters”
Line 16: please delete (FAD) or Fish Aggregating Device, since it is repetitive and it has been already define
Line 17: please replace with widespread
Line 20: please correct into “and used to aggregate pelagic fish species” or add what is missing
Introduction
Line 37: please, replace with FADs
Lines 50-51: please, replace with ‘vulnerable, protected and commercial species, as well as on marine ecosystems’, as FADs is affecting more directly species and indirectly ecosystems
Line 88: please provide replace rules with standards as you refer them (see line 66)
good quality, just few correction as indicated in the comments and suggestions for authors
Reviewer 2 Report
The authors have been accomplished the issues highlighted in the previous version, and now the manuscript is ready to be accepted and published.
Reviewer 3 Report
I want to congratulate the authors for their work on this manuscript, I think it has improved and is ready to continue the publication process.
Some English editing is still needed (e.g. THE Mediterranean, in the title)
